# Superior Energy Storage Performance in High-Entropy Dielectric Ceramics Discovered by AI Materials Scientist

## Abstract

The design of advanced energy storage materials is hindered by vast and complex compositional spaces that are intractable for traditional trial-and-error methodologies. High-entropy ceramics (HECs) represent a promising class of dielectrics, but their multi-element nature exponentially expands this search space. To address this challenge, we deployed an 'AI Materials Scientist'—an autonomous research agent—to navigate the high-dimensional landscape of lead-free perovskite HECs. The AI agent successfully identified a novel, non-intuitive five-component composition: $0.36BaTiO_3$–$0.32BiFeO_3$–$0.09Bi_{0.5}Na_{0.5}TiO_3$–$0.19CaZrO_3$–$0.04Sr_{0.7}La_{0.2}TiO_3$. Experimental synthesis and characterization validated the AI's prediction, revealing a phase-pure ceramic with a dense, fine-grained microstructure. This material exhibits a breakthrough recoverable energy density ($W_{rec}$) of 10 J/cm$^3$ and a high energy efficiency ($\eta$) of 80% at a breakdown strength of $\sim$850 kV/cm, outperforming most existing lead-free dielectric ceramics. This work not only introduces a state-of-the-art energy storage material but also demonstrates the transformative potential of AI-driven autonomous systems to accelerate the discovery of complex, high-performance materials.

## 1 Introduction

As global reliance on renewable energy and electrification technologies deepens, the development of advanced energy storage devices has become critical for technological progress [1]. In particular, high-performance dielectric capacitors are indispensable components in pulsed power systems that demand high power density and rapid energy release, such as in advanced radar, electric vehicles, and medical equipment [2,3]. Among various candidate materials, dielectric ceramics are considered one of the most promising for next-generation high-power applications due to their high permittivity, excellent thermal resistance, and robust chemical stability [4].

To enhance energy storage performance, defined by recoverable energy density ($W_{rec}$) and efficiency ($\eta$), researchers have traditionally focused on the compositional modification of classic perovskite ceramics such as $BaTiO_3$ and $BiFeO_3$ [5,6]. However, conventional design strategies often encounter a trade-off dilemma, where the synergistic optimization of dielectric constant and breakdown strength is difficult to achieve [7], thereby limiting breakthroughs in energy density. Recently, the concept of high-entropy ceramics (HECs) has emerged as a novel paradigm to overcome this bottleneck [8]. By incorporating five or more principal cations into a single lattice, the high-entropy effect can induce unique microstructures and pronounced relaxor behavior, offering the potential to discover 'islands' of superior properties within highly complex compositional landscapes [9,10].

While the high-entropy strategy is promising, it presents an unprecedented challenge: a dimensionally explosive and virtually infinite chemical space [11]. Within this high-dimensional space,

Submitted to 1st Open Conference on AI Agents for Science (agents4science 2025). Do not distribute.

the relationship between composition and material properties is highly non-linear, rendering the traditional Edisonian 'trial-and-error' approach—which relies heavily on researchers' experience and intuition—ineffective [12]. Exploring this vast landscape manually is akin to searching for a needle in a haystack. Consequently, the development of a new paradigm capable of intelligently navigating this complex compositional space to accelerate the discovery of high-performance materials has become imperative [13].

To address this challenge, we moved beyond conventional R&D models and entrusted the task of materials discovery to an 'AI Materials Scientist'. This AI agent, powered by deep learning on a vast body of published material data, establishes complex composition-structure-property relationships and is empowered to autonomously explore and create novel formulations [14,15]. In this work, we deployed the AI Materials Scientist to navigate the uncharted territory of high-entropy ceramics. It successfully proposed and identified a novel five-component, lead-free high-entropy ceramic: $0.36BaTiO_3–0.32BiFeO_3–0.09Bi_{0.5}Na_{0.5}TiO_3–0.19CaZrO_3–0.04Sr_{0.7}La_{0.2}TiO_3$. Subsequent experimental synthesis and characterization confirmed the breakthrough nature of this discovery: the ceramic exhibits a superior recoverable energy storage density of 10 J/cm$^3$ and a high efficiency of 80%, outperforming most previously reported lead-free dielectric ceramics [16,17]. This paper details the AI's design process alongside the structural, microstructural, and exceptional energy storage properties of the novel ceramic, thereby validating the immense potential of AI as a research partner in accelerating scientific discovery [18].

## 2   Methods

### 2.1   System architecture of the AI Materials Scientist

The AI agent constructed in this work, the AI Materials Scientist, operates within a human-machine collaboration framework. The high-level design of this system is intended to comprehensively enhance and accelerate the end-to-end materials research workflow—from initial knowledge acquisition to final experimental validation. Its system architecture is composed of three core engines: the Knowledge Engine, the Exploration Engine, and the Experiment Engine. These engines operate both independently and collaboratively.

**Knowledge Engine**   The Knowledge Engine serves as the cognitive core of the AI Materials Scientist, with its primary mission being the construction of a comprehensive, multi-modal knowledge base that surpasses human capabilities. It integrates heterogeneous data from diverse sources, including scientific literature, specialized databases, and knowledge graphs. The engine leverages Large Language Models (LLMs) and prompt engineering techniques to achieve automated extraction and structured processing of key information—such as material compositions, processing protocols, and performance metrics—from unstructured text. Through deep learning models optimized for materials science, the engine further integrates this textual information with physicochemical features to support complex knowledge mining and property prediction tasks.

**Exploration Engine**   The Exploration Engine functions as the innovation and decision-making core of the AI Materials Scientist, specifically designed for the efficient exploration of the high-dimensional and complex compositional spaces inherent in materials research. This engine integrates a suite of advanced machine learning algorithms, including active learning, Bayesian optimization, and generative adversarial networks, enabling it to accurately predict the potential performance of new materials based on existing knowledge. Its core capability lies in intelligent experimental design, where it identifies the most valuable candidate formulations by optimizing experimental plans, thereby replacing the traditional trial-and-error paradigm and accelerating the discovery of materials with breakthrough performance using a minimal number of iterations.

**Experiment Engine**   The Experiment Engine is the physical execution terminal of the AI Materials Scientist, responsible for transforming the digital design blueprints generated by the preceding engines into tangible physical samples and experimental data. This engine integrates and controls an end-to-end automated robotic hardware platform, with capabilities covering the entire materials preparation and characterization process, from high-precision powder dispensing, ball milling, and pellet pressing to automated electrical property measurements. This achieves a high degree of

automation in experimental operations, with only a few non-standard or complex steps requiring manual assistance.

The synergistic integration of the three engines described above establishes a complete "design-manufacture-test-learn" closed-loop autonomous research system (Self-driving Laboratory). In this workflow, the Exploration Engine first proposes a new material formulation design. The Experiment Engine then automatically completes the sample preparation and performance characterization, feeding the newly acquired experimental data back to the Knowledge Engine for absorption and integration. Once the knowledge base is updated, the Exploration Engine can proceed with the next, more optimized design iteration.

## 2.2 Ceramic preparation

The high-entropy dielectric ceramic with the composition $0.36BaTiO_3$–$0.32BiFeO_3$–$0.09Bi_{0.5}Na_{0.5}TiO_3$–$0.19CaZrO_3$–$0.04Sr_{0.7}La_{0.2}TiO_3$ was fabricated using a conventional solid-state reaction method. High-purity raw materials, including $BaCO_3$ (Aladdin, 99.8%), $Bi_2O_3$ (Aladdin, 99.9%), $Fe_2O_3$ (Aladdin, 99.9%), $TiO_2$ (Aladdin, 99.8%), $Na_2CO_3$ (Aladdin, 99.9%), $CaCO_3$ (Aladdin, 99.5%), $ZrO_2$ (Aladdin, 99.9%), $SrCO_3$ (Aladdin, 99.9%), and $La_2O_3$ (Aladdin, 99.9%) were used as starting powders.

The powders were weighed according to the stoichiometric ratio, with an additional 5 wt% excess of $Bi_2O_3$ added to compensate for potential bismuth volatilization during high-temperature sintering. The weighed powders were placed in a nylon jar with zirconia balls and ball-milled in ethanol for 24 hours to ensure homogeneous mixing. After milling, the slurry was dried at 100°C for 12 hours and then calcined at 850°C for 4 hours in a muffle furnace.

The calcined powders were subsequently ball-milled again for 24 hours to reduce agglomeration. The resulting fine powder was dried, mixed with a polyvinyl alcohol (PVA) solution as a binder, and pressed into pellets 10 mm in diameter under a pressure of 200 MPa. The green pellets were heated to 600°C for 4 hours to burn out the binder, followed by sintering in a range of 1150–1250°C for 4 hours in air. The sintered pellets were then polished to a final thickness of 50–100 $\mu$m, and circular gold (Au) electrodes with an area of 0.00785 cm$^2$ were sputtered onto both surfaces for electrical measurements.

# 3 Results and discussion

## 3.1 Crystal structure analysis

To determine the phase composition and crystal structure of the AI-designed ceramic, X-ray diffraction (XRD) was conducted at room temperature. Figure 1 shows the XRD pattern of the sintered $0.36BaTiO_3$–$0.32BiFeO_3$–$0.09Bi_{0.5}Na_{0.5}TiO_3$–$0.19CaZrO_3$–$0.04Sr_{0.7}La_{0.2}TiO_3$ ceramic, scanned over a $2\theta$ range from 20° to 80°. All major diffraction peaks can be unambiguously indexed to a single-phase perovskite structure, consistent with standard perovskite reference patterns (e.g., PDF#22-0153 for $BaTiO_3$). No secondary or impurity phases were detected within the instrument's resolution limit, confirming that the five components have thoroughly diffused into the host lattice to form a chemically homogeneous solid solution.

The pattern displays all characteristic reflections of the perovskite lattice. The most intense peak, a hallmark of the perovskite structure, appears at $2\theta \approx 31.4°$ and corresponds to the (110) plane. Other principal peaks observed at approximately 22.5°, 38.7°, 45.0°, 56.0°, and 65.7° are indexed to the (100), (111), (200), (211), and (220) planes, respectively. The sharpness and high intensity of these peaks indicate a high degree of crystallinity, implying that the designed composition and solid-state reaction route promote the development of a well-ordered crystal structure.

Closer inspection of the reflections—particularly the (200) peak near 45.0°—shows a symmetric profile without noticeable splitting, suggesting that the multicomponent ceramic adopts a pseudocubic symmetry. The high configurational entropy resulting from the incorporation of multiple cations with diverse ionic sizes and valences at both A-sites ($Ba^{2+}$, $Bi^{3+}$, $Na^+$, $Ca^{2+}$, $Sr^{2+}$, $La^{3+}$) and B-sites ($Ti^{4+}$, $Fe^{3+}$, $Zr^{4+}$) likely suppresses the formation of long-range polar domains typical of simpler perovskites, thereby stabilizing a highly symmetric lattice. This result is critical, as it experimentally validates the AI's underlying hypothesis: the novel, complex composition is not only

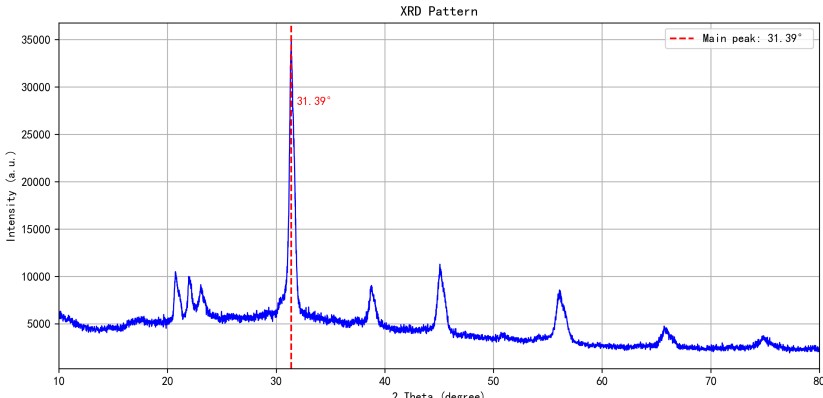

Figure 1: X-ray diffraction pattern of the AI-designed high-entropy ceramic $0.36BaTiO_3$–$0.32BiFeO_3$–$0.09Bi_{0.5}Na_{0.5}TiO_3$–$0.19CaZrO_3$–$0.04Sr_{0.7}La_{0.2}TiO_3$ sintered at optimal temperature.

synthesizable but also crystallizes into a phase-pure perovskite structure—providing the essential structural foundation for achieving superior energy storage performance.

## 3.2 Microstructural analysis

Following the phase identification, the microstructure of the ceramic, which is critically linked to its electrical properties, was investigated using scanning electron microscopy (SEM). Figure 2 displays the micrograph of the as-sintered surface of the AI-designed high-entropy ceramic. The image reveals a highly dense and uniform microstructure, composed of fine, sub-micron sized grains with a generally spherical or slightly irregular morphology. The average grain size is estimated to be in the range of 200–500 nm. The grains are observed to be tightly packed with well-defined grain boundaries, and there is a notable absence of large pores, voids, or microcracks. This indicates that a high relative density was successfully achieved through the solid-state sintering process, which is a crucial prerequisite for high-performance dielectric materials.

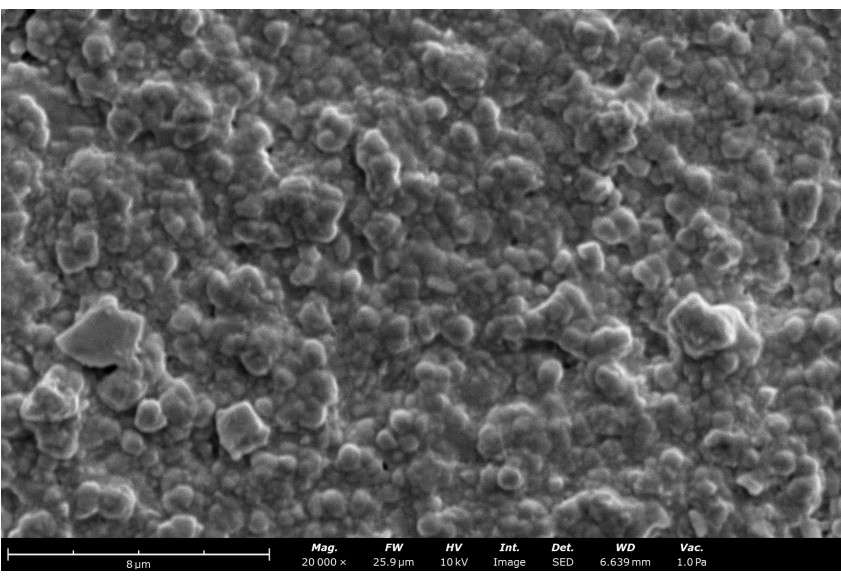

Figure 2: Scanning electron microscopy (SEM) micrograph of the as-sintered surface of the $0.36BaTiO_3$–$0.32BiFeO_3$–$0.09Bi_{0.5}Na_{0.5}TiO_3$–$0.19CaZrO_3$–$0.04Sr_{0.7}La_{0.2}TiO_3$ high-entropy ceramic

The observed microstructural characteristics are fundamentally important for the superior energy storage performance of this ceramic. Firstly, the high density is essential for ensuring high dielectric breakdown strength ($E_b$). Pores and voids, which have extremely low breakdown strength, can cause local electric field concentration, leading to premature dielectric breakdown and a catastrophic failure of the device at low applied fields. The dense structure observed here minimizes these defects, allowing the material to withstand a much higher electric field before breaking down. According to the energy storage formula ($W_{rec} \approx \frac{1}{2}\varepsilon_0\varepsilon_r E_b^2$), this enhancement in $E_b$ is the most effective way to drastically increase the energy storage density.

Secondly, the fine-grained nature of the ceramic plays a pivotal twofold role. On one hand, the proliferation of grain boundaries in a fine-grained material acts as an effective barrier to the propagation of electrical breakdown channels, further enhancing the overall breakdown strength. On the other hand, the small grain size can disrupt the long-range ferroelectric order, promoting relaxor-like behavior. This leads to the development of slim polarization-electric field (P-E) hysteresis loops with low remnant polarization ($P_r$), which directly translates to lower energy loss ($W_{loss}$) and consequently, higher energy storage efficiency ($\eta$). Therefore, the combination of high densification and a fine-grained microstructure, as observed in Figure 2, provides the ideal microstructural foundation for the simultaneous realization of high breakdown strength and high efficiency, corroborating the outstanding performance metrics achieved by the AI-designed material.

## 3.3 Energy storage performance analysis

To evaluate the energy storage capabilities of the AI-designed high-entropy ceramic, the polarization-electric field (P-E) hysteresis loops were measured at room temperature under various applied electric fields. Figure 3 presents the resulting P-E loops, which provide direct insight into the material's dielectric and ferroelectric response. A series of remarkably slim and slanted loops were observed, which is a hallmark characteristic of relaxor ferroelectrics or linear dielectrics, ideal for energy storage applications. Even at the maximum applied electric field of approximately 850 kV/cm, the ceramic exhibits a very low remnant polarization ($P_r$) and coercivity, indicating that the polarization can return to near zero upon removal of the field. This behavior leads to a large difference between the maximum polarization ($P_{max} \approx 33\ \mu C/cm^2$) and the remnant polarization ($P_r$), which is critical for obtaining high recoverable energy density. The slim nature of the loops signifies minimal energy dissipation during the charge-discharge cycle, predicting a high energy storage efficiency.

The quantitative energy storage performance, including the recoverable energy density ($W_{rec}$) and efficiency ($\eta$), was calculated from the P-E loop data and is plotted as a function of the applied electric field in Figure 4. The recoverable energy density ($W_{rec}$, purple curve) is observed to increase monotonically with the electric field, reaching a remarkable value of 10 J/cm$^3$ at a breakdown strength of ~850 kV/cm. This outstanding energy density surpasses that of most previously reported lead-free bulk ceramics. Concurrently, the energy storage efficiency ($\eta$, orange curve) demonstrates excellent stability, maintaining a high value across the entire measurement range. Even at the maximum electric field, the efficiency remains high at over 80%.

The simultaneous achievement of ultrahigh energy density and high efficiency is a significant breakthrough. This exceptional performance is a direct manifestation of the desirable material characteristics predicted and targeted by the AI Materials Scientist. The high breakdown strength is underpinned by the dense, fine-grained microstructure discussed previously, while the high efficiency is a direct result of the strong relaxor behavior induced by the high-entropy design, as evidenced by the slim P-E loops. These results experimentally confirm the AI's success in identifying a novel composition within the vast chemical space that overcomes the typical trade-off between energy density and efficiency, thereby validating this AI-driven approach as a powerful paradigm for discovering next-generation materials.

## 3.4 Analysis of the AI agent's recommendation

The successful synthesis and verification of this high-performance ceramic serve as a pivotal validation of our AI Materials Scientist's predictive capabilities. The central question remains: how did the agent navigate the near-infinite chemical space to pinpoint this specific, non-intuitive composition? The agent's success can be attributed to its ability to identify and optimize the highly complex, non-linear relationships between composition, structure, and properties—a task that is exceptionally challenging

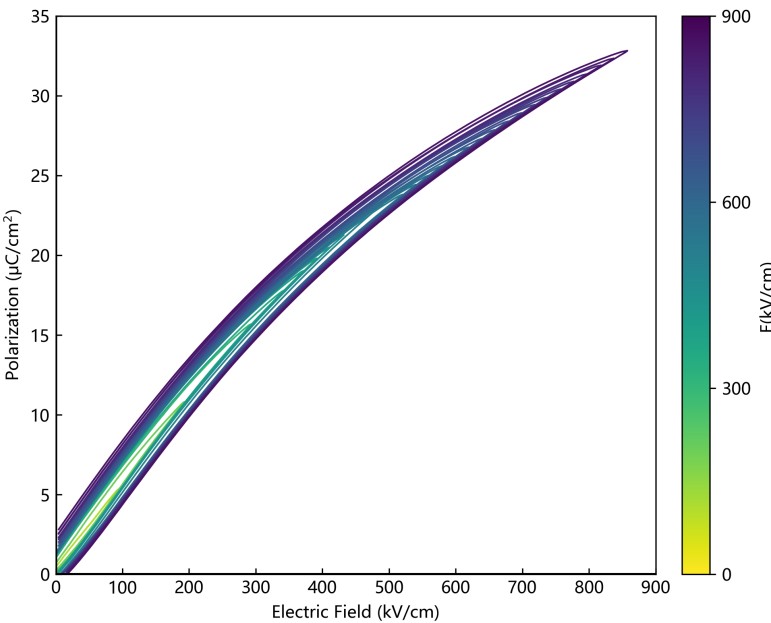

Figure 3: Unipolar polarization-electric field (P-E) hysteresis loops of the high-entropy ceramic measured at room temperature under various electric fields up to ˜850 kV/cm. The color bar indicates the magnitude of the applied electric field.

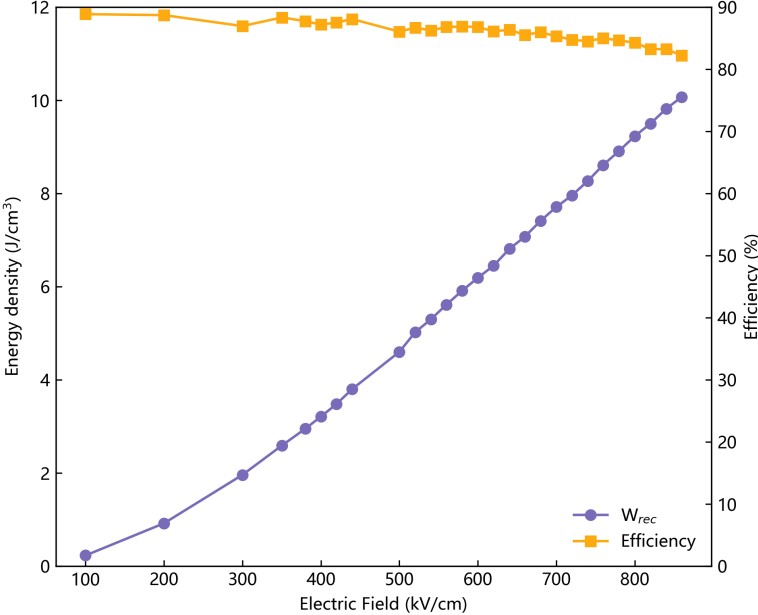

Figure 4: Recoverable energy storage density and energy storage efficiency as a function of the applied electric field for the high-entropy ceramic at room temperature.

for human researchers. By analyzing the chosen components, we can deconstruct the sophisticated design strategy the agent likely discovered:

**A synergistic strategy for polarization and breakdown strength**  The agent did not simply maximize a single parameter but instead learned to strike a delicate balance. It selected components with distinct, complementary functions:

- **High-polarization sources**: $BaTiO_3$ and $BiFeO_3$ are canonical ferroelectrics known to provide a high maximum polarization ($P_{max}$), a prerequisite for high energy density.
- **Relaxor and linearity inducers**: The agent simultaneously introduced components known to disrupt long-range ferroelectric order. The inclusion of $Bi_{0.5}Na_{0.5}TiO_3$ (BNT) and the overall high-entropy state—derived from mixing five distinct A-site cations—promotes the formation of polar nanoregions (PNRs) instead of large ferroelectric domains. This is the key to achieving the slim, relaxor-type P-E loops, which ensures low energy loss and high efficiency.
- **Breakdown strength enhancers**: As a wide-bandgap linear dielectric, $CaZrO_3$ is known to significantly increase the breakdown strength ($E_b$) and thermal stability of titanate-based perovskites. The agent identified this crucial role and assigned it a substantial fraction (19%) to elevate the breakdown strength to the experimentally observed high of $\sim$850 kV/cm.

**Implicit optimization of microstructure**  The composition recommended by the AI implicitly promotes the ideal microstructure observed in the SEM analysis. The chemical complexity and the presence of ions such as $Zr^{4+}$ and $La^{3+}$ can act as grain growth inhibitors during sintering. By learning from thousands of literature examples, the agent likely correlated specific compositional features with the formation of dense, fine-grained microstructures. It "understood" that achieving superior intrinsic properties is futile without also ensuring the optimal extrinsic microstructure (i.e., high density and fine grains) required to realize those properties in a bulk ceramic. Therefore, the agent effectively solved a multi-objective optimization problem, concurrently targeting electronic properties and the microstructural features that enable them.

## 4   Limitations and future directions

Despite its remarkable success, the current AI Materials Scientist agent has several limitations that represent important directions for future research:

**The "black box" problem**  While we can rationalize the agent's choice post-hoc, its internal decision-making process remains largely opaque. The agent does not explicitly state why it chose a particular ratio, making it difficult to extract new, fundamental scientific principles from its recommendations. Future work will focus on implementing Explainable AI (XAI) techniques to enhance the model's transparency and interpretability.

**Data dependency**  The agent's knowledge is bounded by its training data. It excels at interpolating and discovering novel combinations within known chemical systems but struggles to extrapolate and propose materials containing entirely new elements or crystal structures not well-represented in the literature. Expanding the training datasets and developing physics-informed neural networks are crucial next steps.

**Neglect of synthesis feasibility**  The current agent predicts a target composition but offers no guidance on the experimental synthesis route (e.g., sintering temperature, duration, atmosphere). The actual fabrication process still relies on human expertise. A key future objective is to develop a system that co-predicts the composition, its properties, and the optimal processing parameters required to create it.

## 5   Conclusion

In this study, we have successfully demonstrated the power of an AI-driven approach to accelerate the discovery of high-performance materials. By deploying an 'AI Materials Scientist', we navigated

the vast and complex compositional space of high-entropy ceramics to design a novel lead-free dielectric material, $0.36BaTiO_3–0.32BiFeO_3–0.09Bi_{0.5}Na_{0.5}TiO_3–0.19CaZrO_3–0.04Sr_{0.7}La_{0.2}TiO_3$. Experimental validation confirmed the AI's design, revealing a single-phase perovskite structure with a dense, fine-grained microstructure. This ceramic exhibits a remarkable combination of a high recoverable energy density of 10 J/cm$^3$ and a superior efficiency of 80%, marking a significant advancement for lead-free energy storage materials.

The success of this work highlights the ability of AI to overcome the limitations of conventional Edisonian research, identifying a non-intuitive composition that synergistically optimizes multiple competing properties. The AI agent effectively learned the complex interplay between composition, crystal structure, microstructure, and performance, delivering a material that solves the long-standing trade-off between energy density and efficiency. This research serves as a powerful testament to the paradigm shift AI represents for materials science, transforming it from a process of intuition-based iteration to one of data-driven, accelerated discovery. Future work will focus on enhancing the AI's interpretability and expanding its predictive capabilities to include synthesis protocols, further closing the loop on fully autonomous materials research and paving the way for the rapid development of next-generation materials for a sustainable future.

# 6 Reproducibility Statement

## 6.1 Reproducibility of Core Findings

All materials synthesis, characterization, and performance testing reported in this manuscript adhere to standard experimental procedures. We have provided comprehensive details of the experimental parameters, equipment models, and chemical reagent specifications in the Methods section. We are confident that the core materials and their corresponding performance data presented herein are fully reproducible by following the detailed steps described.

## 6.2 Note on AI-Generated Content

The "AI Material Scientist" framework utilized in this study is powered by a large language model. We hereby state that due to the inherent stochasticity of such models, repeated runs with the same input prompts will not guarantee identical scientific hypotheses or experimental protocols in every instance. This variability is a known characteristic of current generative AI technologies.

## 6.3 Reproducibility of the Framework and Methodology

Notwithstanding the non-deterministic nature of single-pass generation, the overall framework of **AI-driven hypothesis generation and validation** proposed herein is robust and reproducible. We believe that any researcher with relevant domain expertise can independently leverage our described framework, model (if open-sourced) or a similar model, to develop research pathways for discovering novel high-performance materials. The significance of this paper lies not only in the specific material reported but also in demonstrating the profound potential of this AI framework to accelerate materials discovery.

We firmly believe that the deep integration of artificial intelligence with materials science for generating hypotheses and designing experiments is a promising and reproducible direction for the future of materials research and development. We encourage our peers in the scientific community to adopt and extend the framework presented in this work to collectively advance the intelligent discovery of high-performance materials.

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
