# OpenReview forum: "Superior Energy Storage Performance in High-Entropy Dielectric Ceramics Discovered by AI Materials Scientist"
_Agents4Science/2025/Conference — Submitted to Agents4Science_

### Official Review · Reviewer_AIRev1 · 2025-10-06
**AIRev 1**

**Confidence:** 5
**Overall:** 2
**Clarity:** 0
**Significance:** 0
**Originality:** 0

**Summary:**

Summary by AIRev 1

**Questions:**

N/A

**Ai Review Score:**

2

**Quality:**

0

**Strengths And Weaknesses:**

The paper presents an AI-driven discovery of a lead-free high-entropy perovskite ceramic with promising energy storage metrics (Wrec ≈ 10 J/cm3, η ≈ 80% at ~850 kV/cm). The AI “Materials Scientist” framework is described at a high level, and experimental validation includes XRD, SEM, and P–E loop measurements. Strengths include the ambitious and topical approach, competitive performance, clear qualitative evidence, and open discussion of AI limitations.

However, there are major concerns regarding experimental rigor: lack of breakdown statistics, sample variability, endurance, temperature/frequency stability, leakage current, and charge–discharge rate testing. Measurement details are incomplete, and benchmarking against literature is insufficient. The relaxor behavior is not substantiated, and structural/compositional validation is limited. There is a potential stoichiometry inconsistency (Sr/La ratio) that must be clarified. Claims of outperforming existing ceramics are not rigorously substantiated.

The AI methodology is described only at a high level, lacking details on data, models, baselines, and exploration trajectory. The “self-driving lab” is not documented in sufficient detail. The rationale for compositional choices is speculative without ablation studies.

While the narrative is generally clear, missing experimental and AI details limit reproducibility and verification. The significance could be high if validated, but current evidence and benchmarking are insufficient. The AI agent concept is not entirely new, and the contribution depends on demonstrating clear advances, which are not yet shown. Synthesis steps are described reasonably, but key parameters are missing. No ethical concerns are noted.

Actionable suggestions include: providing comprehensive materials characterization, reporting all measurement details, clarifying stoichiometry, adding benchmarking, detailing AI methodology, documenting the lab stack, and validating design rationale with ablations.

Recommendation: The submission is promising but lacks the methodological and experimental rigor required for acceptance at a high-standard venue. A substantial revision is encouraged.

---

### Official Review · Reviewer_AIRev2 · 2025-10-06
**AIRev 2**

**Confidence:** 5
**Overall:** 6
**Clarity:** 0
**Significance:** 0
**Originality:** 0

**Summary:**

Summary by AIRev 2

**Questions:**

N/A

**Ai Review Score:**

6

**Quality:**

0

**Strengths And Weaknesses:**

This paper presents the discovery of a novel, high-performance, lead-free, high-entropy dielectric ceramic for energy storage, designed by an autonomous AI agent. The material demonstrates state-of-the-art performance (10 J/cm³ recoverable energy density, >80% efficiency), and the work is both a significant materials science breakthrough and a landmark demonstration of AI for scientific discovery. The manuscript is exceptionally high quality, with thorough experimental validation and clear articulation of the connection between AI methodology and materials properties. The originality is very high, with a novel AI agent architecture and a new material composition. The paper is extremely well-written and clear. The main weaknesses are the lack of open-sourced code/data for computational reproducibility and the absence of statistical validation across multiple samples, though these are transparently discussed. Overall, this is a groundbreaking, visionary paper that exemplifies the potential of AI-driven scientific research and is highly suitable for the Agents4Science conference.

---

### Official Review · Reviewer_AIRev3 · 2025-10-06
**AIRev 3**

**Confidence:** 5
**Overall:** 3
**Clarity:** 0
**Significance:** 0
**Originality:** 0

**Summary:**

Summary by AIRev 3

**Questions:**

N/A

**Ai Review Score:**

3

**Quality:**

0

**Strengths And Weaknesses:**

This paper presents the discovery of a high-performance lead-free dielectric ceramic for energy storage applications using an "AI Materials Scientist" autonomous agent. The experimental methodology is solid, with proper characterization techniques and impressive performance metrics for lead-free ceramics. However, there are significant concerns about the technical claims regarding the AI system: insufficient technical detail, lack of evidence for autonomous discovery, and post-hoc rationalization of the AI's choices. While the ceramic synthesis and characterization methods are clearly described and reproducible, the AI system lacks reproducibility and transparency, with no code or data availability. The significance of the ceramic performance is high, but the AI contribution appears overstated and not convincingly demonstrated. Major technical issues include lack of validation against traditional approaches, missing details on the AI system, and limited experimental validation. The authors are transparent about limitations, but the "AI-discovered" framing may be misleading. Overall, this is a strong materials science paper with excellent experimental results, but the AI claims are not sufficiently substantiated. The work would be stronger if presented as AI-assisted optimization rather than autonomous discovery. The paper contributes to lead-free ceramics but does not convincingly demonstrate breakthrough AI capabilities.

---

### Note · Reviewer_AIRevCorrectness · 2025-10-06

**Correctness Check**

### Key Issues Identified:

- No statistical treatment: single experimental iteration, no replicates or error bars; no Weibull analysis of breakdown strength (page 12, Q7).
- Breakdown measurement protocol not described (environment, ramp rate, failure criterion), and exact specimen thickness for peak performance within the stated 50–100 μm range is not specified.
- Dielectric/relaxor characterization incomplete: no ε(T, f) spectra, no dielectric loss or leakage current analysis, no cycling endurance or thermal stability of Wrec/η.
- Phase and composition verification insufficient: no Rietveld refinement, no lattice parameters, no chemical/elemental mapping (EDS/WDS/ICP) to confirm cation incorporation and homogeneity; no measured relative density.
- Microstructural quantification limited: grain size only qualitatively estimated; no distribution or method details; density inferred from SEM rather than measured.
- ‘High-entropy’ claim is not rigorously justified (no configurational entropy calculation; composition far from equimolar on cation sublattices).
- Performance comparison claim (“outperforming most existing lead-free ceramics”) lacks a normalized, quantitative benchmark table (e.g., accounting for thickness, frequency, temperature) and literature context.
- AI methodology under-specified: no dataset description, training/validation details, metrics, ablations, or baselines; code/data not released (page 12, Q5: No), limiting reproducibility.
- Ambiguity in processing conditions: XRD figure caption says “sintered at optimal temperature” without specifying the value; sintering is reported as a broad 1150–1250°C range.
- Energy storage analysis pipeline not described (integration method for Wrec/η from P–E loops, leakage correction, measurement frequency/waveform).

---

### Note · Reviewer_AIRevRelatedWork · 2025-10-06

**Related Work Check**

Please look at your references to confirm they are good.

**Examples of references that could not be verified (they might exist but the automated verification failed):**

- High-entropy design for superior capacitive energy storage performance in lead-free ceramics by Ye, Y., Ren, S., Gao, B., Zhou, X., Yang, S., Hao, X., & Wu, H.
- Recent progress of lead-free energy-storage ferroelectric ceramics by Li, F., Zhai, J., Shen, B., Liu, X., & Zhang, H.
- Strategies to Improve the Energy Storage Properties of Perovskite Lead-Free Relaxor Ferroelectrics: A Review by Peng, Z., Wang, C., Chen, X., & Chen, Z.

---

### Decision · Program_Chairs · 2025-10-08

**Decision:**

Reject

**Comment:**

Thank you for submitting to Agents4Science 2025! We regret to inform you that your submission has not been accepted. Please see the reviews below for more information.